# OpenReview forum: "EnsembleSHAP: Faithful and Certifiably Robust Attribution for Random Subspace Method"
_ICLR.cc/2026/Conference — ICLR 2026 Poster_

### Official Review · Reviewer_wd8j · 2025-10-29

**Soundness:** 2
**Presentation:** 3
**Contribution:** 2
**Rating:** 4
**Confidence:** 3

**Summary:**

The paper studies explanations for models built with the random subspace method. It proposes an attribution method that reuses random subspace method inference byproducts to produce importance and develops a certified lower bound for detecting explanation-preserving attacks. Experiments focus mainly on text classification and LLM jailbreak detection across different metrics.

**Strengths:**

1. The paper is well organized and easy to follow.
2. The certified lower bound for detecting explanation-preserving attacks provides a certified guarantee that links attribution to robustness.
3. It studies attribution under adversarial, backdoor, and jailbreak settings, which is a novel perspective.

**Weaknesses:**

1. The proposed method is proposed for only random subspace methods, which limits its generalizability to other ensemble methods or broader applications.

2. The experiments emphasize a small set of attribution methods (e.g., Shapley/LIME/ICL). It is suggested to include more diverse families, e.g., propagation-based methods (LRP/DeepLIFT) or gradient-based variants (IG + smoothing), to strengthen the empirical case and reduce experiment bias.

3. Section 5.2.3 shows hyperparameter sensitivity. The analysis mainly compares with Shapley. More attribution baselines and detailed analysis would strengthen this study.

**Questions:**

1. Can the method work beyond RSM?
2. How does your method compare to other attribution families?
3. What causes the hyperparameter sensitivity, and how can it be mitigated?

---

> ### Author Response · Authors · 2025-11-17
>
> We thank the reviewer for the constructive comments.
>
> **Weakness 1: The proposed method is proposed for only random subspace methods, which limits its generalizability to other ensemble methods or broader applications.**
>
> Thanks for the question. As noted by reviewer DnnL, random subspace methods have broad applicability. They are applied to certified robustness at both training time (e.g., poisoning attacks) and inference time (e.g., sparse attacks across different modalities). Beyond robustness, the random subspace method can also be applied to differential privacy (Liu et al., 2020) and machine unlearning (Wang et al., 2024). Exploring how EnsembleSHAP can be extended to these broader applications of random subspace methods is an exciting direction for future research.
>
> **Weakness 2: The experiments emphasize a small set of attribution methods (e.g., Shapley/LIME/ICL).**
>
> Thanks for the suggestion. We also compare our method with DeepLift, a propagation-based approach for the image domain. We use DeepExplainer—an enhanced implementation of DeepLift—on the CIFAR-10 dataset using a CNN classifier, where it has been shown to be effective. Following common practice for random subspace methods (Levine et al., 2019), we use 10,000 random samples. Each pixel is treated as an individual feature, and we report the label flip rate as a function of the percentage of the most important pixels masked. Our method significantly outperforms DeepLift, suggesting that neuron activation information from the base model does not reliably capture the behavior of the ensemble model. We will include this comparison in the camera-ready version.
>
> | Removal ratio | 20%   | 40%   | 60%|
> |---------------|-------|-------|-------|
> | DeepLift | 0.36  | 0.48  | 0.52  |
> | Ours          | **0.82**  | **0.84**  | **0.90**|
>
> We also compare with  Kernel SHAP and ContextCite (state-of-the-art kernel-based methods), and other efficient variants of Shapley, including Beta-Shapley, L-Shapley, and C-Shapley, under default settings. We report faithfulness when removing 20\% of the most important words when there is no attack. The results below show that our method outperforms these baselines, as they cannot guarantee faithfulness to the ensemble model.
>
> | **Dataset**      |  SST-2| |     IMDb||  AG-news | |
> |--------------|-------------|------------|--------------|-------------|------------|--------------|
> | **Removal ratio**      |  **10%**|**20%** |  **10%**|**20%**|**10%** |**20%**|
> | Kernel SHAP    |0.28|0.66 |   0.18| 0.28    | 0.22 | 0.38|
> | ContextCite   |   0.30  | **0.68**|   0.56    | 0.60|0.22|0.38|
> | Beta-Shapley |   0.28  |0.62 |     0.34|0.42   |   0.20   | 0.38
> | L-Shapley    |    0.30 |0.54|    0.38    |0.44 |   0.20  | 0.36
> | C-Shapley    |    0.30   | 0.52      |0.36| 0.42 |0.20| 0.34     |
> | Ours         |  **0.32**  |**0.68**  |  **0.60**  |**0.70** | **0.24** | **0.42**|
>
>
>
> **Weakness 3: Hyperparameter sensitivity analysis should be conducted for more baselines.**
>
> Thank you for the suggestion. In the following tables, we also perform sensitivity analysis on the other two baselines (LIME and ICL) for SST-2. The metric is faithfulness. Our method consistently outperforms the baselines. We will include a more comprehensive sensitivity analysis in the camera-ready version.
>
> | Method | Ours   | LIME   | ICL|
> |---------------|-------|-------|-------|
> | $\rho$=0.6|**0.66**|0.12 |0.12|
> | $\rho$=0.7|**0.68**  | 0.14|0.14|
> | $\rho$=0.8 |**0.68**  |0.16 |0.18|
> | $\rho$=0.9   |**0.44** |0.14|0.18|
>
> | Method | Ours   | LIME   | ICL|
> |---------------|-------|-------|-------|
> | $N$=100|**0.60**|0.20 |   0.16|
> | $N$=1000|**0.68**  | 0.16|0.18|
> | $N$=10000|**0.68** |0.14 |0.18|
>
> **Question 1: Can the method work beyond RSM?"**
>
> While our method is specifically tailored to RSM, RSM serves as a broadly applicable technique in multiple domains, as noted earlier.
>
> **Question 2: How does your method compare to other attribution families?**
>
> As shown in our response to Weakness 2, our method also outperforms attribution methods from other attribution families.
>
> **Questions 3: What causes the hyperparameter sensitivity, and how can it be mitigated?**
>
> The hyperparameter sensitivity primarily arises from how the sampling number $N$ and the dropping rate $\rho$ influence the utility of the ensemble model and the robustness of the explanation. We note that both the robustness and utility of the method improve as the sampling number $N$ increases. Thus, we recommend choosing a relatively large $N$ whenever computational resources permit. For the dropping rate $\rho$, we observe a trade-off between utility and robustness. In practice, setting $\rho=0.8$ offers a good balance between the two.
>
> **References**
> 1. Liu et al. (2020) On the Intrinsic Differential Privacy of Bagging
> 2. Wang et al. (2024) Machine Unlearning: A Comprehensive Survey
> 3. Levine et al. (2019). Robustness Certificates for Sparse Adversarial Attacks by Randomized Ablation

---

> > ### Author Response · Authors · 2025-12-01
> >
> > Dear reviewer,
> >
> > Thank you for your valuable feedback.  As the discussion period is ending soon, we wanted to kindly highlight our clarifications above.  Any additional comments from your side would be sincerely appreciated.
> >
> > Thanks again for your time.

---

### Official Review · Reviewer_DnnL · 2025-10-30

**Soundness:** 3
**Presentation:** 2
**Contribution:** 3
**Rating:** 8
**Confidence:** 4

**Summary:**

This work proposes an enhancement for arbitrary partition-based certified defenses that (1) provides explanability and (2) is in itself certifiably robust.

For context, partition-based defenses, here referred to as "random subspace methods", are a commonly used flavor of randomized smoothing that performs self-ensembling over (random) subsets of features or training samples. If the number of subsets that a single element can appear in is bounded, then it can only have limited influence on the ensemble prediction. This allows for the derivation of robustness certificates.

The authors propose a method for computing per-feature importance scores given the ensemble model's prediction.
They then show that this method has three desirable properties:
* The importance scores can be computed from the same feature subsets that are already used for making smoothed predictions. Thus, the computational overhead is small.
* The importance scores inherit desirable properties of Shapley values, which are a powerful (but in this setting intractable) feature attribution method
* Using the standard argument for partition-based defenses, the method is in itself certifiably robust. Specifically, if an adversarial attack on the ensemble model is successful, then many of the modified features will be assigned a high importance score.

In the experiments, the method is first empirically evaluated based on its ability to provide useful (as measured by "faithfulness") explanations under adversarial attacks on standard classification datasets, as well as an LLM jailbreaking benchmark. Afterwards, the certificates are evaluated in terms of certified detection rate (essentially certified ratio for explanations).

**Strengths:**

* The proposed method is very broadly applicable, with partition-based defenses being a standard approach to certified robustness at both training time (poisoning attacks) and inference time (sparse attacks on images, graphs, point clouds etc.)
* The approach is incredibly elegant, achieving multiple goals via relatively straight-forward procedure (see desirable properties above)
* Section 4.3 provides provable utility guarantees in addition to provable robustness guarantees. This is somewhat unusual for randomized smoothing papers and definitely a positive
* The chosen range of datasets and models appears adequate for a paper that is primarily focused on provable robustness
* Provably robust explanation is mostly underexplored, i.e., novelty appears high

**Weaknesses:**

The work is primarily held back by its presentation. In particular:
* There are various typos and grammatical errors (tense, conjugation, duplicate words like "is a is a" in l.075 etc.). The manuscript could be significantly improved by running it through grammarly, the Copilot grammar checker, or a similar tool
* The main theorem is somewhat awkwardly forwarded (see Eq. 12-16) and incredibly dense. If the theorem itself cannot be further simpliified, I would encourage the authors to at least expand its intuitive explanation in l. 332ff.
* There are no explanatory figures. Adding one or two could help readers not familiar with randomized smoothing to more easily follow the paper.

Other than that, I only see two issues with the experimental evaluation:
* When varying parameters of the explanation method (see, e.g., Fig. 16), only the effect on certified detection rate is shown. However, it is not clear how varying these parameters impacts model utility. It would be better to additionally visualize the trade-off between utility and provable robustness (similar to certified accuracy in classification).
* Assuming the authors do, in fact, only want to show certificate strength: Constraining the experiments to a specific dataset and model as in Fig. 1 leads to a reductive view on the certification procedure (it fixes the dataset size and the model's confidence etc. to a particular value). It would be more informative to just treat these as additional parameters to vary (as is already done with $\beta$, $\rho$, $N$, etc.). Varying these additional parameters could be a nice experiment for the camera-ready version.

### Summary
Overall, this work makes a novel, elegant, and broadly applicable contribution to the field of provably robust machine learning There is
 minor room for improvement in the experimental evaluation.
Assuming that the authors will improve the presentation for the camera-ready version (at least fixing most of the grammatical errors and typos), I recommend acceptance.

**Questions:**

### Other suggestions:

The following paper appeared ca. three months before the submission deadline. I would encourage the authors to discuss it as concurrent work:

Anani et al.. Pixel-level Certified Explanations via Randomized Smoothing. ICML 2025

---

> ### Author Response · Authors · 2025-11-17
>
> We thank the reviewer for the constructive suggestions.
>
> **Weakness 1. Presentation issues**
>
> Thank you for the suggestions. We will refine the grammar, correct the typos, and include a figure illustrating the random subspace method in the camera-ready version. We will also add more explanation to help readers parse Equations 12–16 and provide additional intuition behind the derivation of the theorem.
>
> **Weakness 2. Utility and provable robustness trade-off**
>
> Thank you for the suggestion. The table below reports how utility (accuracy of the ensemble model) varies with different dropping rates $\rho$ and sample numbers $N$. Figures 16 and 17 in the paper appendix present the corresponding changes in explanation robustness. We observe a trade-off for the dropping rate $\rho$: increasing $\rho$ improves explanation robustness but reduces utility. In contrast, increasing the sample number $N$ enhances both utility and robustness. We will incorporate these results into the paper.
>
> | **Dataset** |  SST-2|      IMDb|  AG-news |
> |--------------|-------------|------------|--------------|
> | $\rho$=0.6    |0.80|0.84 |   0.96|
> | $\rho$=0.7   |   0.80  | 0.84|   0.96    |
> | $\rho$=0.8 |   0.76  |0.80 |     0.94|
> | $\rho$=0.9   |    0.68 |0.76|    0.90    |
>
> | **Dataset** |  SST-2|      IMDb|  AG-news |
> |--------------|-------------|------------|--------------|
> | $N$=100    |0.70|0.78 |   0.92|
> | $N$=1000  |   0.76  | 0.80|   0.94    |
> | $N$=10000 |   0.80  |0.82 |     0.94|
>
>
>
>
> **Weakness 3. Could vary additional parameters (e.g.,model, dataset)**
>
> Thanks for the constructive suggestion. Figures 15, 16, and 17 in the Appendix show the effect of several hyperparameters (e.g., confidence level) across three different datasets. In the camera-ready version, we will further expand our evaluation by incorporating additional models/datasets for a more comprehensive view on the certification strength.
>
> **Suggestion 1. Discuss related work**
>
> Thanks for the suggestion. Our paper focuses on explanation-preserving attack, where an attacker can adversarially cause the classifier to misclassify while hiding the perturbed features from the explanation. Related theoretical works on robust feature attribution (Anani et al. 2025; Wang et al., 2024; Lin et al.,2023; Li et al.,2023; Chen et al.,2019; Wang et al., 2020; Wang et al., 2023) focus on prediction-preserving attacks, where the adversary aims to drastically change the explanation while generally keeping the classifier’s prediction fixed. We will add discussion and references into the paper.
>
> **References**
> 1. Anani et al. (2025). Pixel-level Certified Explanations via Randomized Smoothing
> 2. Wang et al. (2024). Certified l2 Attribution Robustness via Uniformly Smoothed Attributions
> 3. Lin et al. (2023). On the Robustness of Removal-Based Feature Attributions
> 4. Li et al. (2023). Robust Data Valuation with Weighted Banzhaf Values
> 5. Chen et al. (2019). Robust Attribution Regularization
> 6. Wang et al. (2020). Smoothed Geometry for Robust Attribution
> 7. Wang et al. (2023). Certification of Attribution Robustness for Euclidean Distance and Cosine Similarity Measure

---

> ### Comment · Reviewer_DnnL · 2025-11-17
>
> Thank you for your rebuttal.
>
> Concerning Weakness 3:
> I am sorry, it seems like I did not express my suggestion properly:
> What I meant is that, ultimately, the dataset and model *do not matter* in partition-based certificates. In partition-based defenses, the model only influences the class probabilities plugged into the certificate. The dataset only influences the number of partitions at a given partition size, which is again some scalar parameter plugged into the certificate.
>  So instead of running these experiments for many different models and datasets, one could evaluate the certificate (albeit not the robustness-accuracy trade-off) by varying over class probabilities and dataset sizes directly (and save a lot of compute in the meanwhile).
>
> Either way, I still think that this is a good paper and should definitely be accepted. The other listed weaknesses have been adequately addressed.

---

> > ### Author Response · Authors · 2025-11-17
> >
> > Thank you for the clarification. We now fully understand your suggestion: you recommend directly varying the scalar parameters that appear in Theorem 1—such as class probabilities and partition sizes—in addition to evaluating across multiple models and datasets. This would make the results more general, model and dataset-independent, while also reducing computational burden for evaluation. We agree that this is a meaningful and insightful evaluation. We will add this analysis into the camera-ready version.
> >
> > Thank you again for your positive assessment and confidence in our work.

---

### Official Review · Reviewer_QMf6 · 2025-11-01

**Soundness:** 2
**Presentation:** 3
**Contribution:** 2
**Rating:** 2
**Confidence:** 4

**Summary:**

This paper proposes EnsembleSHAP, an explanation method for Random Subspace Method ensembles that is efficient and certifiable. During the inference of an ensemble method, EnsembleSHAP uses the model’s prediction to assign credit to features, then derives a certified detection lower bound of explanation-preserving attacks that modify at most T features. Experiments show that EnsembleSHAP achieves higher faithfulness with small overhead.

**Strengths:**

1.	The method suits well with RSM since it explains the ensemble by reusing its votes, so it is efficient and well-motivated.
2.	The method is model agnostic since it only requires votes and some ablations, making it easy to adapt.
3.	Various types of attacks are evaluated such as jailbreaking, token edits, trigger based.

**Weaknesses:**

1.	The certification is limited to at most T feature perturbations, which may be limited since many real-life attacks can modify an arbitrary number of features (e.g., sinusoidal signal poisoning).
2.	The tools used in theoretical analysis are fairly elementary, such as frequency counting, confidence intervals, and binary search, so the novelty of the theorems and proofs appears limited.
3.	The threat model is limited for explanation-preserving attacks only, which is overly restrictive: in practice, many attackers will at least partially disrupt the explanation (e.g., by shifting saliency), so this limited certification guarantee seems limited to real-world scenarios.

**Questions:**

1.	This method assumes equal contribution among a subset of features. What is the rationale behind this choice?

---

> ### Author Response · Authors · 2025-11-17
>
> We thank the reviewer for the thoughtful comments.
>
> **Weakness 1. The certification is limited to at most T feature perturbations, which may be limited since many real-life attacks can modify an arbitrary number of features (e.g., sinusoidal signal poisoning).**
>
> In this work, we focus on $l_0$-certification scenario (Levine et al.,2020; Chiang et al.,2020; Levine et al.,2019; Zeng et al., 2021), where he attacker can arbitrarily modify a bounded number of features. We leave the exploration of other certification paradigms as future works.
>
> **Weakness 2. The tools used in theoretical analysis are fairly elementary, such as frequency counting, confidence intervals, and binary search, so the novelty of the theorems and proofs appears limited.**
>
> Our work provides the first theoretical framework for certified robustness against explanation-preserving attacks. The key novelty lies not in using complex tools, but in offering a new conceptual angle that makes certification possible in the first place. Our main theorem establishes a provable lower bound on the certified detection size—the minimum possible overlap between adversarial features and the model-reported important features, conditioned on the attack being successful.
>
> To derive this result, we introduce a new proof strategy grounded in the law of contraposition. Rather than directly bounding the detection size—something that appears intractable—we take an unconventional approach: we instead ask “What must happen if fewer than $k$ adversarial features are detected?” By applying contraposition, we convert this into a certifiable guarantee: whenever the reverse of this inequality holds, the detection size must be at least $k$. This contraposition-based reasoning, combined with our certified detection formulation, constitutes the core novelty of our theoretical contribution and distinguishes it from prior work.
>
> **Weakness 3. The threat model is limited for explanation-preserving attacks only, which is overly restrictive: in practice, many attackers will at least partially disrupt the explanation (e.g., by shifting saliency), so this limited certification guarantee seems limited to real-world scenarios.**
>
> We apologize for the confusion. Although we adopt the term “explanation-preserving” to stay consistent with prior work, our threat model does not actually restrict the diruption of the explanation. In our setting, the attacker is able to arbitrarily perturb a bounded number of features to induce misclassification—these perturbations can also arbitrarily disrupt the explanation. The attacker’s objective is to ensure that the perturbed features still receive low importance scores from the explanation method, which makes such attacks difficult to detect.
>
> **Question 1. This method assumes equal contribution among a subset of features. What is the rationale behind this choice?**
>
> Sorry for the confusion. An accurate description is that, given a subsampled feature group, the contribution of each feature to this group’s result is regarded as the same. The rationale is to ensure that the symmetry axiom (from the Shapley value framework) is satisfied: if two features always make the same marginal contribution across all possible feature groups, then they must receive the same importance score.
>
> **References**
> 1. Levine et al. (2020). (De)Randomized Smoothing for Certifiable Defense against Patch Attacks
> 2. Chiang et al. (2020). Certified Defenses for Adversarial Patches
> 3. Levine et al. (2019). Robustness Certificates for Sparse Adversarial Attacks by Randomized Ablation
> 4. Zeng et al. (2021) Certified Robustness to Text Adversarial Attacks by Randomized [MASK]

---

> ### Author Response · Authors · 2025-12-01
>
> Dear reviewer, we really appreciate your earlier comments.  We have now provided detailed responses to all questions.  If possible, we would be grateful for any further feedback you might offer.
>
> Thanks again for your time.

---

### Meta-Review · Area_Chair_VeXZ · 2026-01-06

**Summary:**

Reviewers mainly pointed out the following concerns:
- Limited scope focusing on RSMs (wd8j and QMf6) and explanation-preserving attacks (QMf6) -- also assuming equal contribution of features
- Limited evaluations against diverse attribution baselines (wd8j)
- Missing analysis of utility-robustness trade-off (DnnL)
- Relative simplicity of the analysis (QMf6)

**Reviewer Concerns:**

Authors convincingly resolved concerns over the limitation to explanation-preserving attacks by pointing out the contrapositive providing a certificate when detection fails, as the perturbed features cannot stay hidden.  Assuming equal contribution is needed for the symmetry axiom of Shapley values.  The limitation to RSMs is mildly addressed citing its use in differential privacy and machine unlearning.

The authors provided significant additional experiments during the rebuttal phase, addressing concerns over limited baselines for evaluation (notably showing strong improvements over DeepLift, highlighting where existing methods fall short) and also assessing hyperparameter sensitivity.

The authors acknowledge the theoretical derivations may rely on simpler tools, but the conceptual merit stands.

Overall, the breadth of applicability afforded by an efficient approach as demonstrated by experiments on different types of attacks (backdoor, adversarial, and jailbreak) - readily demonstrate the value of the contribution and the potential for subsequent developments.

**Reviewer Scores:**

Initial scores came as 8/4/2.  Following the discussion, with substantial clarification over limitations and significant new results, I would expect a final score above 6.

---

### Decision · Program_Chairs · 2026-01-26

Accept (Poster)